# Quality by Design Based Formulation of Xanthohumol Loaded Solid Lipid Nanoparticles with Improved Bioavailability and Anticancer Effect against PC-3 Cells

**DOI:** 10.3390/pharmaceutics14112403

**Published:** 2022-11-07

**Authors:** Vancha Harish, Devesh Tewari, Sharfuddin Mohd, Pilli Govindaiah, Malakapogu Ravindra Babu, Rajesh Kumar, Monica Gulati, Kuppusamy Gowthamarajan, SubbaRao V. Madhunapantula, Dinesh Kumar Chellappan, Gaurav Gupta, Kamal Dua, Siva Dallavalasa, Sachin Kumar Singh

**Affiliations:** 1School of Pharmaceutical Sciences, Lovely Professional University, Jalandhar-Delhi G.T Road, Phagwara 144411, India; 2Department of Pharmacognosy and Phytochemistry, School of Pharmaceutical Sciences, Delhi Pharmaceutical Sciences and Research University, New Delhi 110017, India; 3Department of Pathology, School of Medicine, Wayne State University, Detroit, MI 48202, USA; 4Faculty of Health, Australian Research Centre in Complementary & Integrative Medicine, University of Technology Sydney, Ultimo, NSW 2007, Australia; 5Department of Pharmaceutics, JSS College of Pharmacy, JSS Academy of Higher Education & Research, Ooty 643001, India; 6Center of Excellence in Molecular Biology and Regenerative Medicine Laboratory (A DST-FIST Supported Center), Department of Biochemistry (A DST-FIST Supported Department), JSS Medical College, JSS Academy of Higher Education and Research, Bannimantapa, Sri Shivarathreeshwara Nagar, Mysore 570015, India; 7School of Pharmacy, International Medical University, Bukit Jalil, Kuala Lumpur 57000, Malaysia; 8School of Pharmacy, Suresh Gyan Vihar University, Mahal Road, Jaipur 302017, India; 9Department of Pharmacology, Saveetha Dental College and Hospitals, Saveetha Institute of Medical and Technical Sciences, Saveetha University, Chennai 602105, India; 10Uttaranchal Institute of Pharmaceutical Sciences, Uttaranchal University, Dehradun 248007, India; 11Discipline of Pharmacy, Graduate School of Health, University of Technology Sydney, Ultimo, NSW 2007, Australia

**Keywords:** Box–Behnken Design, oral bioavailability, drug release, solid lipid nanoparticles, Xanthohumol

## Abstract

Many natural products with greater therapeutic efficacy are limited to target several chronic diseases such as cancer, diabetes, and neurodegenerative diseases. Among the natural products from hops, i.e., Xanthohumol (XH), is a prenylated chalcone. The present research work focuses on the enhancement of the poor oral bioavailability and weak pharmacokinetic profile of XH. We exemplified the development of a Xanthohumol-loaded solid lipid nanoparticles (XH-SLNs) cargo system to overcome the limitations associated with its bioavailability. The XH-SLNs were prepared by a high-shear homogenization/ultrasonication method and graphical, numerical optimization was performed by using Box–Behnken Design. Optimized XH-SLNs showed PS (108.60 nm), PDI (0.22), ZP (−12.70 mV), %EE (80.20%) and an amorphous nature that was confirmed by DSC and PXRD. FE-SEM and HRTEM revealed the spherical morphology of XH-SLNs. The results of release studies were found to be 9.40% in 12 h for naive XH, whereas only 28.42% of XH was released from XH-SLNs. The slow release of drugs may be due to immobilization of XH in the lipid matrix. In vivo pharmacokinetic study was performed for the developed XH-SLNs to verify the enhancement in the bioavailability of XH than naive XH. The enhancement in the bioavailability of the XH was confirmed from an increase in C_max_ (1.07-folds), AUC_0-t_ (4.70-folds), t_1/2_ (6.47-folds) and MRT (6.13-folds) after loading into SLNs. The relative bioavailability of XH loaded in SLNs and naive XH was found to be 4791% and 20.80%, respectively. The cytotoxicity study of naive XH, XH-SLNs were performed using PC-3 cell lines by taking camptothecin as positive control. The results of cytotoxicity study revealed that XH-SLNs showed good cell inhibition in a sustained pattern. This work successfully demonstrated formulation of XH-SLNs with sustained release profile and improved oral bioavailability of XH with good anticancer properties against PC-3 cells.

## 1. Introduction

Tremendous interest has emerged towards Xanthohumol (XH), a naturally occurring bioactive prenylated chalcone from hops with various pharmacological activities. Prenylflavonoids, which are from hops, have various biological activities against many ailments, including neoplasms, osteoporosis, postmenopausal hot flashes, digestive issues, neuralgia, toothaches, tension headaches, and earaches. Chemically, XH is written as 30-[3,3-dimethyl allyl]-20,40,4-trihydroxy-60-methoxychalcone (Figure 1). XH’s structure was identified first in 1957 by [1], but its advantageous pharmacological properties were not treasured until the 1990s. In 2007, the Committee on Herbal Medicinal Products of the European Medicines Agency reported XH use in conventional medicine for the gentle treatment of symptoms of insomnia and mental stress. Further, the treatment for sleep disturbances, anxiety, and some other diseases by hops has been approved by Commission E of the Germany and European Scientific Cooperative on Phytotherapy [2].

Even though XH has extensive pharmacological effects, its delivery to the target site is challenging because of its high hydrophobic nature, and poor oral bioavailability (pharmacokinetic profile). This is mainly due to its biotransformation in the gastro-intestinal tract, by hepatic enzymes and accumulation of 70% XH in apical side of Caco2 cells. About 93% of the intracellular XH is localized in the cytosol by binding to the cytosolic proteins [3]. Therefore, XH fails to produce the effective therapeutic response at the target site. As a result of these limitations of XH, it is required to develop the formulation that renders all the limitations associated with XH oral bioavailability and transport to the target site without losing at the biotransformation sites. Novel drug delivery systems have opened new avenues to transport the biological molecules to the target site by rendering the key obstacles associated with their transportation.

Solid lipid nanoparticles (SLNs) are one of the promising nanocarriers in medical care, because of their various advantages such as stability, biocompatibility, non-toxicity, controlled release and drug targeting over other nanocarriers. The particle diameter of SLNs ranges from 50–1000 nm and also exhibits high cellular uptake [4,5]. They have been extensively reported for delivery of bioactive natural compounds to mainly treat diseases including cancer [6], obesity [7], diabetes [8] and neurodegenerative diseases [9]. The SLNs based formulations have been reported to markedly enhance the bioavailability and physical stability of drugs such as carvediol [10], efavirenz [11], famotidine [12], resveratrol [13] and pomegranate extract [14]. In SLNs, the movement of incorporated drug molecules becomes restricted, which protects the leakage of the drug from the carrier [15]. Both hydrophilic and hydrophobic drugs can be incorporated into SLNs based on the type of method selected for preparation [16]. SLNs are composed of biodegradable solid lipids/mixture of lipids, surfactants (stabilize the structure of SLN), co-surfactants (optional), aqueous phase, solvents/co-solvents, charge modifiers, stealthing agents (for improving long circulation on time and targeting ability of SLN), cryoprotectants and active pharmaceutical ingredients (Drugs, proteins, DNA, plasmid and genes). The drug is loaded into the lipid matrix, which is coated by the surfactant and is in solid form at room and body temperatures. Due to the high biocompatibility of solid lipids, SLNs can be administered through all routes [17]. Many researchers have demonstrated successful loading of various anticancer agents into SLNs and evaluated them against prostate cancer cell lines. Beg et al., 2022, systematically developed SLNs loaded with abiraterone acetate to improve the oral biopharmaceutical properties and for the treatment of prostate cancer. The loading of abiraterone acetate into SLNs have successfully improved the oral bioavailability and anticancer activity against PC-3 cell lines, which was confined by in vitro and in vivo studies [18]. Jalilian et al., 2021, developed targeted SLNs for docetaxel with anisamide as ligand to treat prostate cancer. The outcomes of the research have shown that docetaxel loaded SLNs with anisamide has acted more effectively on PC-3 and HEK293 cell lines than free drug and docetaxel-loaded SLNs without anisamide [19]. Akanda et al., 2015, developed SLNs loaded with retinoic acid for delivery and treatment of prostate cancer. The prostate cancer effect was evaluated by using LNCap human prostate cancer cells [20]. Oner et al., 2021, has developed cationic SLNs to carry siRNAs to target EphA2 receptor, which is over-expressed in prostate cancer. Therefore, Oner’s research has demonstrated that SLNs can also be used to deliver siRNA [21]. SLNs can be prepared by several methods such as hot and cold high-pressure homogenization, solvent evaporation method, solvent injection, microemulsion, membrane contactor methods, hot and cold high shear homogenization/ultrasonication, super critical technology, phase inversion temperature method, coacervation method, double emulsion method, emulsification–solvent evaporation method, emulsification–solvent diffusion method, and electrospray method [22].

In the present study, an attempt has been made to develop and optimize XH-loaded SLNs (XH-SLNs) using a quality by design approach for improving the bioavailability and to overcome the limitations associated with pure XH. Homogenization and ultrasonication method have been used to develop formulation.

## 2. Materials and Methods

### 2.1. Materials Used

XH (Xantho Flav) was gifted by Simon H. Steiner, Hopfen, GmbH (Mainburg, Germany). Compritol E ATO (CE), Precirol ATO5 was gifted by Gattefosse, (Mumbai, India). Lipoid LIPOID E 80SN (LE-80), Lipoid S 75, and Phospholipon 90 H were gifted by Lipoid GmbH (Germany). Pluronic F-68 and tween 80 were purchased from Hi-Media laboratories (Mumbai, India). Monemul-20 was gifted from Mohini chemicals, Mumbai, India. Glyceryl monostearate (GMS) and carnauba wax were purchased from Sigma-Aldrich (New Delhi, India). Palmitic acid and cetyl alcohol were purchased from Lobachemie Pvt Ltd. (Mumbai, India). Stearic acid was purchased from Qualigen fine chemicals (Mumbai, India). Sephadex G-25, dialysis bag (14–15 kDa) was purchased from GE Healthcare, Hyderabad. Methanol and orthophosphoric acid (OPA) of HPLC grade were purchased from Molychem (Mumbai, India). Water was passed through Milli Q filter to prepare the mobile phase.

### 2.2. Methodology

#### 2.2.1. Reverse Phase-High Performance Liquid Chromatography Analysis of XH

XH was estimated by using previously developed and validated RP-HPLC method by using Shimadzu HPLC system (Japan) equipped with mobile phase delivery system (LC-20 AD), photodiode array detector (SPDM20A) and rheodyne sample injecting loop (20 μL). LC solution software was used as data station for gathering information related to analytes. Methanol and 0.1% *v*/*v* orthophosphoric acid (pH 1.9) [10:90% *v*/*v*] as mobile phase, 0.8 mL/min as flow rate, and injection volume of 20 µL. The chromatogram was observed at a wavelength of 370 nm. The validation data is mentioned in Appendix A under method validation and results of method validation sections.

#### 2.2.2. Determination of XH Solubility in Various Solid Lipids

Solubility studies were done for solid lipids like GMS, stearic acid, palmitic acid, carnauba wax, CE, Precirol ATO5 and Cetyl alcohol were used for solubility study. Initially, 10 mg of lipid was taken in a beaker of 10 mL capacity. The beaker was then placed on a magnetic stirrer and the lipids were melted by heating them at 10 °C above their melting point. XH (10 mg) was accurately weighed and added to the molten lipid with constant stirring using a magnetic micro bead on the magnetic stirrer (REMI, Mumbai, India). The temperature of the stirrer was maintained at 100 °C [23,24]. The molten lipid was added to the beaker until a clear solution was achieved. Total amount of lipid added to form clear solution was noted (Table 1). The solid lipid, which was able to dissolve the maximum amount of the drug in a lower quantity of lipid, was selected for further study. 

#### 2.2.3. Development of XH-SLNs

XH-SLNs were prepared by homogenization–ultrasonication method. Briefly, the lipophilic phase was prepared by heating the solid lipid 10 °C above the melting point of the lipid. Drug and lipophilic surfactant were added to the molten lipid and mixed. Secondly, the aqueous phase was prepared by dissolving hydrophilic surfactant in double distilled water. After reaching the equilibrium temperature, the aqueous phase was dispersed in the lipophilic phase drop-wise through burette using a homogenizer (REMI) at 8000 rpm. The resultant XH-SLN dispersions were sonicated for 10 min using probe sonicator (Lab India) and cooled to room temperature for solidification of nanoparticles [24,25,26]. 

#### 2.2.4. Selection of Surfactant and Co-Surfactant

Appropriate surfactant and co-surfactant were selected by preparing XH-SLNs by the method described in Section 2.2.2. by taking an individual fixed amount of CE (60 mg) and surfactant, co-surfactant (2% *w*/*v*) as well as their mixture in the ratio of 2:1 (such as Pluronic F-68, Tween80, Monemul-20, LIPOID S 75, LE-80 and PHOSPHOLIPON 90 H). The resultant XH-SLNs were analyzed for PS, PDI and ZP. The surfactant and co-surfactant that give small PS, narrow PDI and optimum ZP were selected for optimization [23,27,28].

#### 2.2.5. Design of Experiment

##### Preliminary Screening of Lipid, Surfactant, Co-Surfactant and Time, Speed of Homogenization

The preliminary screening is the most important step in DoE for the selection of critical processing parameters (CPP) and critical manufacturing attributes (CMAs) that affect critical quality attributes (CQA). The screening of solid lipid, surfactant and cosurfactant was performed in order to obtain smaller PS, narrow PDI, optimum ZP and greater entrapment efficiency (%EE) of the SLNs. Drug to lipid ratio and lipids were screened for optimization based on the solubility study and partitioning behavior of the drug into the lipid. Further, lipids with good solubility were taken, and XH-SLNs were prepared using various surfactants/co-surfactants in various concentrations to screen the best lipid, surfactant and co-surfactant combination from which the effects of surfactant/co-surfactant on CQAs were studied. The homogenization speed and time were screened by varying the time and speed of homogenization, according to the procedure mentioned in Section 2.2.3, and evaluated for PS, PDI, ZP, %EE and effect on CQAs without probe sonication.

##### Box–Behnken Design (BBD) for Optimization

Finally, based on the data obtained from preliminary screening, three level, 3^3^ factor design (BBD) was selected for optimization of XH-SLNs. The optimization was done by studying the effect of CMAs such as amount of lipid (A), amount of surfactant (B) and concentration of co-surfactant (C) on CQAs or dependent variables such as PS (R_1_), PDI (R_2_), ZP (R_3_) and %EE (R_4_). CE, LE-80 and PF-68 were selected as lipid, surfactant, co-surfactant, respectively, and were optimized at three levels +1 (high), 0 (medium), −1 (low). Design expert software (Version 11, stat-ease. Inc., Minneapolis, MN, USA) was used to optimize the variables. To obtain the best optimized formulation, a total of 17 experiments were performed based on the design and the responses were recorded. Further, the design has helped in generating a polynomial equation to understand the impact of variables on responses. The best-fitting experimental model (linear, two-factor interaction, quadratic, and cubic) was chosen statistically by comparing several statistical parameters such as coefficient of variation (CV), multiple correlation coefficient (R^2^), adjusted multiple correlation coefficient (adjusted R^2^), predicted residual sum of square, and graphically by 3D response surface plot. The software recommended the model with the highest determination coefficients and significance value at the selected probability. Statistical validation of the generated mathematical polynomial equations was carried out by utilizing the software’s ANOVA provision. A *p*-value of less than 0.05 was judged significant. 

#### 2.2.6. Data Optimization and Model Validation

The optimization of the formulation was done by utilizing graphical (overlay plot) and numerical (desirability) criteria to achieve the desired goal, i.e., smaller particle size, narrow PDI, optimum zeta potential and %EE. Checkpoint analysis was used to determine the accuracy of the established model by comparing the degree of error between observed and predicted values.

#### 2.2.7. Lyophilization of Optimized XH-SLNs

In the present study, a shelf lyophilizer (Esquire biotech, EBT-12N, Chennai, India) with a freezing capacity up to −50 °C was used. Optimized XH-SLNs (20 mL) were placed in a 40 mL wide mouth fast-freeze flask tube and were mixed with the mannitol (cryoprotectant) and pre-frozen for 12 h at −20 °C and lyophilized for 48 h at −30 °C in a freeze dryer and 20 mTorr pressure. Secondary drying was carried out at 20 °C and 5 pascal pressure for 6 h in order to obtain dried XH-SLNs. After the freeze drying process, the vacuum was broken using ambient air and after reaching to the atmospheric conditions, the samples were unloaded from the lyophilizer [29,30,31].

### 2.3. Characterization of Optimized XH-SLNs

Various characterization parameters were performed for the optimized XH-SLNs formulation such as particle size, PDI, zeta potential, %EE, percentage yield, % drug loading, DSC, FTIR, PXRD, FE-SEM, HRTEM, in vitro release studies, drug release kinetic study and pharmacokinetic study.

#### 2.3.1. Determination of Particle Size, PDI and Zeta Potential

Zetasizer Nano ZS90 (Malvern Instruments, Malvern, UK) was used for the determination of PS, PDI and ZP. PS of lyophilized and freshly prepared dispersion of XH-SLNs was determined by diluting up to 10 times in double distilled water. The 1 mL of diluted sample was taken in a disposable cuvette and examined at 25 °C at an angle of 90° using a helium-neon laser as light source, where diffusion of the particle due to Brownian motion was converted into PS. The sample was taken in a disposable folded capillary cell for ZP examination.

#### 2.3.2. Determination of %EE, %DL, and Percentage Yield

The %EE of the XH-SLNs was determined by separating the entrapped and unentrapped XH using Sephadex G-25 as stationary phase. The XH present in XH-SLNs was extracted after lysis of lipid particles by mixing with methanol followed by filtration through 0.22 µm filter. Then, both entrapped and unentrapped XH content was determined in triplicate by using the HPLC method mentioned in Section 2.2.1. Concentration of the XH was calculated by using the calibration curve. %EE, %DL and percentage product yield were calculated by following equations:(1)$$\%\mathrm{EE}=\frac{\mathrm{Total}\;\mathrm{amount}\;\mathrm{of}\;\mathrm{XH}\;-\;\mathrm{Amount}\;\mathrm{of}\;\mathrm{free}\;\mathrm{XH}}{\mathrm{Total}\;\mathrm{amount}\;\mathrm{of}\;\mathrm{XH}}\times 100$$
(2)$$\%\mathrm{DL}=\frac{\mathrm{XH}\;\mathrm{weight}\;\mathrm{in}\;\mathrm{the}\;\mathrm{nanoparticles}}{\mathrm{Total}\;\mathrm{weight}\;\mathrm{of}\;\mathrm{nanoparticles}}\times 100$$
(3)$$\mathrm{Percentage}\;\mathrm{yield}=\frac{\mathrm{Total}\;\mathrm{nanoparaticles}\;\mathrm{weight}}{\mathrm{Total}\;\mathrm{solid}\;\mathrm{weight}}\times 100$$

#### 2.3.3. Assay of XH and pH of XH-SLNs

The formulation containing XH equivalent to 10 mg was diluted 10 times in methanol before being further diluted with mobile phase. XH content was measured using the HPLC after the dilution of samples using the formula given in Equation (4). pH of XH-SLNs formulation was measured at room temperature using a calibrated digital pH meter.
(4)$$\%\mathrm{purity}=\frac{\mathrm{sample}\;\mathrm{area}}{\mathrm{standard}\;\mathrm{area}}\times 100$$

#### 2.3.4. Differential Scanning Calorimetry (DSC)

DSC analysis of XH, CE, Lipoid E SN80, Pluronic F-68, physical mixture, blank SLNs and XH-SLNs was performed by using DSC 6000, Perkin Elmer, Waltham, MA, USA. The analysis was done for studying the crystalline behavior of drug in SLNs. The blank SLNs and XH-SLNs should be lyophilized before DSC study. For melting point and heat of fusion, the device was calibrated using indium (calibration reference 99.9% pure). In the range of 30–300 °C, a heating rate of 10 °C/min was used. The experiment was carried out with purging of nitrogen gas (50 mL/min). Samples (4 mg) were placed in conventional aluminum pans, with an empty pan serving as a control. DSC thermograms of pure XH, solid lipid, physical mixtures and XH-SLNs was recorded.

#### 2.3.5. Powder X-ray Diffraction (PXRD)

Powder X-ray diffraction is an important analytical technique which is used to determine the powder characteristics like, degree of crystallinity, crystal lattice arrangement of the formulation. It also aids in identifying the physical state (crystal or amorphous) of the prepared formulation. The PXRD diffractograms of XH, CE, Lipoid E SN80, Pluronic F-68, SLNs without drug (Blank SLNs), XH loaded SLNs (XH-SLNs) were recorded by using powder X-ray Diffractometer (Bruker D8 Discover PXRD) with a Ni-filtered Cu-K radiation at a voltage of 40 kV and current of 30 mA. Diffractograms were obtained using a step size of temperature 0.045 degrees (θ) and step time 0.5 s with a detector resolution of 2Ɵ diffraction angle between 2° and 60° at room temperature and which are analyzed by JCPDF software of PXRD. 

#### 2.3.6. Field Emission Scanning Electron Microscopy (FESEM)

FESEM (JSM-6610LV), JEOL, Peabody, MA, USA was used to investigate the surface morphology of naive XH, PF-68, LE-80, CE, XH-SLNs, as it gives topographical and elemental information with virtually unlimited depth. Light sprinkling of SLNs was done on a double adhesive carbon tape that was attached to an aluminum stub. It was used to prepare the samples for scanning electron microscopy. The stub was then coated using a gold sputter module in a high vacuum evaporator in an argon environment. After that, the samples were scanned and photomicrographs were made at magnifications of 10–300,000×.

#### 2.3.7. High Resolution Transmission Electron Microscope

The surface characteristics and morphology of XH-SLNs were observed by using HRTEM (JEM2100Plus Electron Microscope, JEOL company, Peabody, MA, USA). A drop of SLNs was stained negatively charged using 1% aqueous solution of phosphotungstic acid. This was placed on a micropipette having 200 micron mesh-size pioloform-coated copper grid. The film was dried for 1 h and analyzed under TEM at 50–80 kV [32].

#### 2.3.8. In Vitro Drug Release

The in vitro drug release from the XH-SLNs was performed using a dialysis bag-based diffusion method. Before using, the bag was soaked in distilled water for 12 h for activation. The suspensions of naive XH and lyophilized SLNs were put into a dialysis bag (HiMedia, Cutoff 14 kDa) that was held with clips. The bags were inserted into a beaker filled with 50 mL of methanolic PBS pH 7.4 (50% *v*/*v*). XH has limited solubility in buffer, but is soluble in methanol, hence methanol was added to PBS pH 7.4 to maintain the perfect sink conditions [33,34,35]. The beakers were covered with aluminum foil to avoid evaporation of methanol during experiment and were placed on a thermostatic magnetic stirrer (Remi, Mumbai) and stirred at 100 rpm and 37 °C [36,37]. The aliquots of 2 mL from the release media were taken out and replaced with the same volume of fresh medium at specified time intervals (0, 2, 4, 6, 8, 12, 18, 24, 30, 36, 48, 60, 72, 92 h). The HPLC technique was used to analyze aliquots that had been filtered using a 0.22 µm nylon syringe filter.

#### 2.3.9. Drug Release Kinetics

The data obtained from in vitro release studies of XH-SLNs were fitted to various kinetic models such as zero order, first order, Higuchi model, Weibull, Hixon Crowell and Korsmeyer Peppas model. The mechanism and kinetics of drug release were determined by the obtained correlation coefficient (R^2^) [37]. The model showing the highest value of R^2^ is considered.

#### 2.3.10. Cell Permeability Study

Permeation of the developed XH-SLNs was evaluated in Caco2 cell monolayer to determine the quick absorption of medication through SLNs. Caco2 cells were cultivated in a 12 mm trans well polycarbonate membrane with 0.4 mm holes for 21days. Before the transcellular evaluation, cells were given a higher transepithelial electrical resistance value of 300 Ω/cm^2^ and rinsed three times with Hanks balanced salt solution (pH 6.5). In order to investigate the permeation, 0.5 mL and 1.5 mL of transport buffers were placed in the side A (apical) and side B (basolateral), respectively. The naive XH solution was prepared by suspending it in 0.5% *w*/*v* carboxy methyl cellulose solution. The XH-SLNs and naive XH solution were added to the side A and side B of the cell inserts, respectively. The 0.1 mL of solution from side A and side B were collected at predetermined time intervals of 1, 2, 3, 4 and 5 h set in advance. The amount of sample collected from the both sides was replaced with the same amount of fresh transport buffer. The drug content in the samples collected were analyzed by using HPLC after filtration. The amount of drug permeated with respect to time was recorded [38].

#### 2.3.11. In Vivo Pharmacokinetic Study

##### Procurement and Storage of Animals

Eighteen male Sprauge Dawley rats were purchased from National Institute of Pharmaceutical Education and Research (NIPER), Mohali, India for the present study. The age of all rats was between 10 and 11 weeks and weight in the range of 250–350 g. The rats were kept in polypropylene cages lined with husk under the temperature of 25 ± 2 °C; relative humidity of 55 ± 10% and 12:12 light: dark cycle. The animals were fed with standard pellet diet and water at libitum. The study protocol was approved by the Institutional Animal Ethics Committee of School of Pharmaceutical Sciences, Lovely professional University (Protocol no: LPU/IAEC/2021/88).

##### Pharmacokinetic Study

A total of 18 rats were used for the pharmacokinetic study. A parallel study was followed, and the rats were divided into three equal groups (group 1, group 2, and group 3). The group 1 rats (n = 6) received naive XH, (dose = (30 mg/kg, p.o.)), group 2 rats (n = 6) received XH-SLNs (dose = (30 mg /kg, p.o.)), group 3 received blank SLNs (placebo). It is pertinent to note that naive XH was used as standard and XH-SLNs were used as test formulations. Naive XH was used as standard in this study. The rats received the naive XH suspended in 0.5% *w*/*w* CMC suspension (group 1) and XH-SLNs (group 2) and placebo were given after reconstitution with water. The formulations were administered to rats after 24 h of fasting. In all the cases, blood samples (0.5 mL) were withdrawn from the tail vein site at 2, 12, 24, 36, 48, 72, 84, 96, 108, and 120 h in vials containing ethylene diamine tetra acetic acid (EDTA) as anticoagulant. The blood samples were mixed well, centrifuged, and the plasma was transferred to 5 mL vials, capped tightly and stored at −40 °C for further analysis. Blood (0.5 mL) was withdrawn from the tail vein from alternate animals of the same group at different time intervals. The pharmacokinetic parameters such as area under curve (AUC), C _max_, T _max_, and T_1/2_ were calculated by using the PK plus module of gastro plus software version 9.2. The relative bioavailability (Fr) was calculated by using Equation (5).
(5)$$\mathrm{Relative}\;\mathrm{bioavailability}\;\left(\mathrm{Fr}\right)=\frac{{\mathrm{AUC}}_{\mathrm{test}}\times {\;D}_{\mathrm{std}}}{{\mathrm{AUC}}_{\mathrm{std}}\times {\;D}_{\mathrm{test}}\;}\times 100$$
where, AUC is Area Under Curve, D is dose administered.

#### 2.3.12. Cytotoxicity Study on PC-3 Cell Lines

The in vitro cell line toxicity for naive drug, XH SLNs, Placebo SLNs and camptothecin (positive control) was performed using PC-3 cell lines using 3-(4,5-dimethylthiazol-2-yl)-2,5-diphenyltetrazolium bromide (MTT) assay. PC-3 cells were plated at a density of 5 × 10^3^ cells per well in 100 mL of Dulbecco’s Modified Eagle Medium (DMEM) containing 10% FBS in 96-well plates and grown for 24 h. Cells were then exposed to a series of samples viz. naive drugs, naive drug, XH SLNs, Placebo SLNs and camptothecin at different concentrations, i.e., 3.125, 6.25, 12.5, 25, 50, and 100 µM concentration for a period of 48 h. The sampling was done at four different time intervals, i.e., 6, 12, 24 and 48 h. Cell viability was determined as a measure of succinate dehydrogenase released by the viable cells, which reduces the tetrazolium salt of MTT into formazan. The percentage cell inhibition was calculated using the formula given below.
(6)$$\%\;\mathrm{cell}\;\mathrm{inhibition}=100-\left[\left(\frac{A_{t}-A_{b}}{A_{c}-A_{b}}\right)\right]\times 100$$
where, A_t_ = Absorbance value of test compound; A_b_ = Absorbance value of blank; and A_c_ = Absorbance value of control.

## 3. Results and Discussion

### 3.1. Determination of XH Solubility in Various Solid Lipids

The solubility of the drug within the solid lipid core has a major impact on the capacity of SLNs to accommodate a particular drug, which has a great impact on the limiting factor, i.e., entrapment of the drug in the solid lipid [39]. Therefore, optimization of the solid lipid is critical because the amount of drug solubilized has a significant influence on the %EE of the drug. The solubility of XH in multiple solid lipids was studied to optimize the %DL and %EE of the developed SLNs, allowing for the selection of a solid lipid that was ideal for producing the SLNs [40]. A total of eight solid lipids were taken for the solubility study, out of which CE has shown maximum solubility than other lipids (Table 1). These findings were confirmed by the improper matrix structure of CE molecules, due to the presence of mono-, di-, triacylglycerols and glycerides that provide loose, highly porous structural properties, allowing for easy drug accommodation and increased solubility. Therefore, CE is selected as a lipid core for the preparation of XH-SLNs [27,41,42,43].

### 3.2. Selection of Surfactant and Co-Surfactant

Surfactant and co-surfactant selection were made based on the capacity of surfactants to give smaller PS, narrow PDI, ZP and non-formation of precipitation. Even though the surfactant has higher XH solubility, they may not have the capability to emulsify the lipid surface. Therefore, selection is based on their emulsification capacity. XH-SLNs were prepared with the selected solid lipid and various surfactants (PF-68, Tween 80, Monemul-20, LIPOID S 75, LE-80, and PHOSPHOLIPON 90 H) and evaluated for PS, PDI, ZP and precipitation. The XH-SLNs formula (Table 2) containing PF-68 and LE-80 have shown least PS (118.20 ± 0.14 nm), narrow PDI (0.17 ± 0.078), and acceptable ZP (−12.60 ± 2.84 mV). In addition to this, the formula does not produce any sign of precipitation. Thus, LE-80 was selected as surfactant and PF-68 was selected as co-surfactant/stabilizer for developing XH-SLNs. Both the selected surfactant and co-surfactant were non-ionic surfactants which are safe for oral delivery and may not produce any irritation in a physiological environment. The PS is greatly dependent on the hydrophilic–lipophilic balance (HLB) of the applied surfactant. Greater HLB values correspond to smaller particles.

### 3.3. Preliminary Screening of Lipid, Surfactant, Co-Surfactant and Time, Speed of Homogenization

Initially, three lipids (GMS, Precirol ATO5 and CE) with relatively higher solubility were selected based on the results obtained from solubility study and XH-SLNs were prepared by the selected method using LE-80 (2%), Pluronic F-68 (2%) (previously selected) as surfactant/co-surfactant and evaluated for PS, PDI, ZP and %EE. Among these solid lipids, CE based SLNs have shown has given smaller size (135.70 ± 3.65 nm), narrow PDI (0.28 ± 0.32), optimum ZP (−12.70 ± 3.13 mV) and higher %EE (74.60 ± 2.74%) than GMS and Precirol ATO5 based SLNs (Table 3). This could be due to the presence of mono-, di-, triglycerides as well as fatty acids of various chain lengths that create less ordered crystals with many lattice imperfections, which aids in accommodation of large amounts of drug. Based on the above findings, CE was selected as solid lipid. LE-80 and PF-68 were used as surfactant and co-surfactant, respectively. It was noticed that there was a significant change in PS, PDI, ZP and %EE upon changing the amount of lipid, amount of LE-80 and concentration of Pluronic F-68. As a result, amounts of CE, LE-80 and concentration of PF-68 were selected as independent variables (CMAs) for optimization of XH-SLNs by Box–Behnken Design (BBD).

The effect of homogenization speed and time on PS, PDI, ZP and %EE was studied by formulating XH-SLNs with the fixed amount of the selected lipid and surfactant/co-surfactant, by varying the homogenization time (15, 20, 30 min) and speed (6000, 8000, 10,000 rpm). It was revealed that, when the speed of homogenization increased from 6000 rpm to 8000 rpm and time of homogenization from 15 min to 20 min, the PS (351.60 ± 3.43 nm to 140.60 ± 4.89 nm) and PDI (0.55 ± 0.08 to 0.26 ± 0.03) were decreased and %EE (70.60 ± 2.15% to 78.50 ± 3.43%) increased. However, when homogenization speed was increased from 8000 rpm to 10,000 rpm and time of homogenization was varied from 20 min to 30 min, size (140.60 ± 4.89 nm to 259.40 ± 4.28 nm) and PDI (0.26 ± 0.03 to 0.68 ± 0.05) were increased and % EE (78.50 ± 3.43% to 68.80 ± 2.76%) significantly decreased (Table 4). This might be attributed to the fact that once a stable lipid core with decreased PS and PDI has been created, increasing kinetic energy has no effect on PS and PDI [24]. The rupturing of lipid core occurs and drug molecules may leak out from the lipid matrix into the external phase due to high kinetic energy that leads to poor drug loading capacity of SLNs. As a result, the speed of homogenization (8000 rpm) and time of homogenization (20 min) were kept as constant values during the optimization of XH-SLNs. Characteristic responses of prepared XH-SLNs such as PS, PDI, ZP and %EE were selected as dependent variables (CQAs).

### 3.4. Box–Behnken Design (BBD) for Optimization

A statistical 3^3^-BBD (Design Expert, version 11; Stat-Ease Inc., Minneapolis, MN, USA) was used for the development and optimization of XH-SLNs and to analyze the main and interaction effects of CMAs [amount of CE (A), amount of LE-80 (B) and concentration of PF-68 (C)) on CQAs (PS (R_1_), PDI (R_2_), ZP (R_3_), and %EE (R_4_)]. BBD has provided a series of 17 randomized runs (XH-SLNs formulae) with five center points. These were determined using factors that were operating at three different levels (+1 (high), 0 (medium), −1 (low)). All the XH-SLNs were prepared as per the suggested design and evaluated for PS, PDI, ZP and %EE (Table 5). The optimization was performed with the observed responses and fitted to linear, 2FI (linear-two factor interaction), quadratic and cubic models. Various polynomial equations were derived for the models to look for their goodness to fit best fit. The software recommended a model with greater correlation coefficients (R^2^) and significance value (*p* < 0.05) at the stated probability level. ANOVA provision available in the software was used for validation of the derived mathematical polynomial equation statistically (Appendix A). Numerical and graphical optimization was performed with the help of statistically designed equations for studying the interactions of CPPs on CQAs by generating two-dimensional (2D) counter plots, and three-dimensional (3D) surface plots. 

Fitting the obtained results of all the dependent variables into various polynomial model equations revealed that the independent variables have linear (R_1_), 2FI (R_2_) and quadratic (R_3_ and R_4_) interaction effects on the observed responses, with enhanced multiple correlations, adjusted as well as the predicted, sum of squares and significant statistical terms at the selected probability level. The fit statistics for all responses clearly indicate that there is a reasonable agreement between the predicted R^2^ and adjusted R^2^ values (difference is <2).

An adequate precision ratio, which measures the signal to noise ratio that aids in navigating the design space, greater than 4 is desirable. In the present study, the adequate precision value was found above 4 for all the responses (R_1_ to R_4_). The polynomial equations were validated by applying ANOVA. The results of ANOVA indicated significant model F-values with *p* < 0.0001 values for all the responses, indicating less signal–noise ratio (0.01% chance of noise) (Appendix A). These findings were consistent with the predicted adequate precision values, indicating that the recommended model equations were suitable for navigating the design space. The 3D-response surface plots were plotted by performing the numerical and graphical optimization to investigate the actual interactive effects of CPPs on CQAs. They mainly help in studying the effect of two CPPs at a time by keeping other CPPs constant on each response. To compare and quantify the influence of a single variable on each response, perturbation plots were created to show how each response behaved when one component changed within the defined constraint range while the other two remained constant. The desirability function for PS is minimum, PDI is minimum, ZP is in range and %EE is maximum level.

#### 3.4.1. Effect of Independent Variables on PS

Several parameters such as the amount of solid lipid, concentration of surfactant/co-surfactant, sonication time and homogenization conditions (speed and time) influence the PS of XH-SLNs [39]. The PS of XH-SLNs ranges from 96.23 ± 3.72 nm for XH-SLN_15_ to 195.40 ± 4.36 nm for XH-SLN_17_ (Table 5). The quantitative interaction effect of CPPs on PS are represented by the following polynomial equation that was found to be linear:(7)PS=+153.64+30.93∗A+12.99∗B+7.03∗C

The values of F > *p* < 0.05 mention that the model terms are significant (Appendix A). The generated polynomial equation is clearly indicating that all the factors (A, B, C) have significant effect on PS. Positive sign before the coefficient indicates the synergistic effect of the factors on the response (PS). Among all the three factors, factor A (amount of CE) has great impact on the PS, i.e., increase in the amount of CE in the XH-SLNs formulation, there is a dramatic increase in the PS (negative impact), which was undesirable and also had chances of hindering the release of the drug from the lipid to outer phase that affects the therapeutic response (Figure 2A and Figure 3A). The results presented in Table 5 show that factor A had a synergistic impact on PS, with XH-SLN_4_, XH-SLN_6_, XH-SLN_16_ and XH-SLN_17_ (A = 60 mg) having considerably higher PS than XH-SLN_2_, XH-SLN_9_, XH-SLN_12_ and XH-SLN_15_ (A = 30 mg) at constant B and C, respectively. This synergistic effect of CE is explained by the enhanced surface tension and consistency of the formulation as the content of CE grows with subsequent PS augmentation. Factors B (amount of LE 80) and C (Concentration of PF-68) also had synergistic effects on the PS as revealed from the polynomial equation and from the data presented in Table 5**.** As the amount of factor B increased (B = 300 mg) in the formula, there is a notable increase in the PS (XH-SLN_3_, XH-SLN_4_, XH-SLN_8_, and XH-SLN_9_), whereas the PS was found less while using 100 mg of factor B (XH-SLN_7_, XH-SLN_13_, XH-SLN_15_ and XH-SLN_16_). This is mainly due the amphiphilic natural phospholipid used as surfactant in XH-SLNs by considering the advantages of LE-80 in oral formulations. Similarly, factor C also has little synergistic impact on PS. An increase in concentration of PF-68 (0.3%) in the formula along with the factor B increased the PS (XH-SLN_8_, XH-SLN_12_, XH-SLN_13_ and XH-SLN_17_) than the formulation containing 0.1% of factor C (XH-SLN_2_, XH-SLN_3_, XH-SLN_6_, and XH-SLN_7_). This effect was explained based on the fact that a greater concentration of both surfactant and co-surfactant lowered surface tension, reduces PS and avoided agglomeration of particles [44,45]. The results of the experiments showed that there was a decrease in the PS with an increase in the concentration of factors B and C up to the optimal ratio (1:2). The increase in the concentration of B and C produced inverse effects, i.e., growth in PS at a constant amount of factor A and C. Lastly, it was confirmed that factor A has a greater impact on PS, whereas factors B and C have negligible impacts.

#### 3.4.2. Effect of Independent Variables on PDI

PDI describes the dispersity of the particles in the dispersion. The results recorded from the experiments showed the PDI values ranged from 0.18 ± 0.03 (XH-SLN_15_) to 0.29 ± 0.06 (XH-SLN_4_). The interactive effects of CPPs on PDI are given by the following 2FI polynomial equation:(8)PDI=+0.22820.0384∗A+0.0135∗B+0.0006∗C+0.0130∗AB+0.0002∗AC+0.0230∗BC

Significant model terms are indicated by the values of F > *p* < 0.05 (Appendix A). According to the polynomial equation, all the factors (A, B, and C) had a synergistic effect on the PDI. Only factor A had greater impact on PDI whereas factor B and C had negligible impact on PDI. It was attributed to the increase in the PS by increase in the amount of factor A that led to aggregation of the particles. This caused enhancement of PDI (0.29 ± 0.06) due to the presence of more CE that provided more space for the drug molecule to embed in it, ultimately leading to minimization of the total surface area.

Based on the observed PDI values and polynomial equation, it was possible to infer that factor B and C had no significant influence on the PDI values of the various formulations examined. Upon increase in the concentration of factor B and C, there was a slight change (increase or decrease) in the PDI. This was mainly because of the decreased interfacial tension between the lipid phase and aqueous phase that controlled the particle aggregation (Figure 2B and Figure 3B). According to earlier studies, a higher surfactant concentration efficiently stabilized the lipid matrix by generating a steric barrier on its surface, thereby preventing aggregation. When the concentration of factor B increased from 100 mg to 300 mg in the formulation, there was a steady increase in PDI. It was attributable to the fact that the alkyl chain of surfactant/co-surfactant encapsulated the surface of the SLNs through hydrophobic connection, resulting in the formation of a stable lipid matrix [27,46]. When this steady framework of lipid was produced, extra surfactant caused deposition of the surfactant particle on the surface of the steady framework. This resulted in an increase in PDI, as observed in the developed formulations [24,26].

#### 3.4.3. Effects of Independent Variables on ZP

ZP is termed as electro-kinetic potential present on the surface of the particles that measures the stability of the colloidal dispersions, including SLNs. ZP hampers the internal phase agglomeration and reaggregation. It also measures the interparticle repulsion inside the colloidal system. The value of ZP of colloidal dispersion mainly depends on the charge present on the particle. The charge on the surface of the particle was due to the surface-active agents and their HLB values. The lower ZP values indicated the instability of the dispersion and also led to coalescence [14,47,48]. In the present study, the surfactant and co-surfactant used were non-ionic in nature. The ZP values for the developed XH-SLNs ranged from −1.98 ± 4.52 mV (XH-SLN_17_) to −11.90 ± 4.52 mV (XH-SLN_11_). The effects of CMAs on ZP were explained by the given quadratic equation:(9)$$\mathrm{ZP}=+11.58+0.4913*A+0.0550*B+3.21*C+0.3350*\mathrm{AB}+0.3725*\mathrm{AC}+0.0250*\mathrm{BC}+0.3862{*A}^{2}+0.8787{*B}^{2}+4.93{*C}^{2}$$

As per the results shown in Table 5 and the quadratic equation, all the factors have a synergistic effect on ZP, but with less impact, which was indicated by the coefficient values in the equation. As the concentration of factors, A, B, and C increased, the value of ZP decreased. Higher values of surfactant and co-surfactant showed the ZP values as −1.98 ± 4.52 mV (XH-SLN_17_), −2.56 ± 3.56 mV (XH-SLN_13_), −3.92 ± 3.12 mV (XH-SLN_12_), and −2.76 ± 3.86 mV (XH-SLN_8_), whereas the combination of medium and lower values showed ZPs of −11.50 ± 1.85 mV (XH-SLN_1_), −11.50 ± 3.25 mV (XH-SLN_5_), −10.60 ± 4.85 mV (XH-SLN_15_), and −10.50 ± 7.25 mV (XH-SLN_16_). These results may be described by the beneficial influence of the amount of CE on PS, which leads to an enhanced surface area of the particle allowing for a higher charge density and higher ZP (Figure 2C and Figure 3C) [14].

#### 3.4.4. Effect of Independent Variables on %EE

The results obtained from the experiments were represented in Table 5. The %EE was ranging from 23.80 ± 1.05% (XH-SLN_6_) to 76.00 ± 1.86% (XH-SLN_16_) based on various factor levels. The quadratic polynomial equation for % EE is given by Equation (10):(10)$$\%\mathrm{EE}=+75.60+12.30*A-11.50*B+11.95*C-12.10*\mathrm{AB}-8.80*\mathrm{AC}+7.70*\mathrm{BC}-15.25{*A}^{2}+1.15{*B}^{2}-19.75{*C}^{2}$$

According to the quadratic equation, factors A, B and C had significant effects on %EE. As per the obtained results, factors A and C had synergistic effects and factor B had an antagonistic effect. Changing each variable separately while keeping the other factors constant resulted in a considerable rise in the proportion of entrapped drugs. A higher coefficient of factor A (+12.30) indicated that the amount of CE was the major factor that affected %EE. The increase in %EE was mainly due to the presence of a higher amount of CE that produces a greater space to accommodate the drug. This phenomenon reduced the movement of the drug to the external phase because of the higher viscosity of the lipid, which increased the %EE. Increase in the concentration of the surfactant caused decreased %EE due to higher solubilization of the drug. As the solubility of XH increased in the external phase, more of the drug could be diffused from the lipid core, which would have led to a decrease in the %EE [49,50]. Increase in the concentration of factor B and C significantly increased the %EE. Increase in the concentration of factor A and C and decrease in the value of factor B (XH-SLN_17_) dramatically decreased %EE of XH. Decrease in the concentration of factor A and B and increase in the concentration of C significantly decreased the %EE of XH. 

### 3.5. Data Optimization and Validation

The critical response contours were superimposed on a contour plot in Design Expert software to perform graphical optimization. This resulted in an overlay plot (Figure 4A) with two regions: yellow indicating an area of a design space with possible response values, and grey describing an area where response values did not meet the quality product criterion [24,51]. The optimum batch was chosen based on the overlay plot and desirability criteria (Figure 4A). It mainly contains three CMAs such as A—amount of CE (30 mg), B—amount of LE-80 (149.691 mg), C—concentration of PF-68 of (0.203% *w*/*v*) and fixed levels of homogenization speed of 8000 rpm and time 20 min, as well as probe sonication time for 10 min at amplitude 40% and pulse rate of 30. The predicted values given by the design for PS, PDI, ZP and %EE were 116.351 nm, 0.189, −11.244 mV and 73.068%, respectively. The observed results for the optimized formula were shown in Figure 4B. The formulations for validation were prepared using the predicted quantities of the CMAs as mentioned above. All CQA responses have shown an R^2^ value of more than 0.9 for the predicted versus observed values and *p*-value more than 0.05, suggesting that there was a level of high agreement between them. The magnitude of error is important for determining the accuracy of the produced equations and expressing the model’s relevant domain [27,51].

### 3.6. Characterization of Optimized XH-SLNs

The characterization parameters like PS, PDI, ZP, %EE, %DL, DSC, FTIR, pH, Assay, and morphological analysis (FE-SEM, HE-TEM) of optimized XH-SLNs were performed.

#### 3.6.1. Determination of PS, PDI and ZP

The PS, PDI and ZP analysis for the optimized XH-SLN was performed in triplicate using Zeta sizer Nano ZS90, Malvern, UK, dynamic light scattering (DLS) and Laser Doppler Electrophoresis. PS of XH-SLNs was found to be 108.60 ± 3.21 nm with PDI 0.22 ± 0.04, which is indicating narrow size distribution (Figure 5). The storage stability of XH-SLNs was described by the ZP that indicates the charge on the SLNs. The ZP value of optimized XH-SLNs was found to be −12.7 mV, which was negative. This is ascribed to the non-ionic character of the surfactants utilized in this study. Despite having a reduced ZP, SLNs were found to be stable due to the presence of non-ionic surfactants that provide steric stability to the system [52,53]. ZP was decreased due to the adsorption of these steric stabilizers that causes significant repulsion between particles, thereby avoiding aggregation during storage.

#### 3.6.2. %EE and %DL

The type of lipid used in the formulation of XH-SLNs plays a crucial role in entrapping XH. The lipid used in development of XH-SLNs was CE, which was a mixture of mono-, di-, and triglycerides. In addition, various long chain fatty acids have an imperfect crystal lattice that allows the XH molecules to accommodate in it. This was evident with the %EE of 80.20 ± 2.95% and %DL of 12.40 ± 1.63% with respect to the amount of drug used. Percentage yield of XH-SLNs after lyophilization was found to be 85.54 ± 2.98%.

#### 3.6.3. Assay and pH

The assay value of 97.68% revealed that a greater quantity of XH was present in the entire SLNs. XH-SLNs showed pH value of 6.65, which was within the normal pH range of 5.8–7.4 of oral cavity. Therefore, it was expected that XH-SLNs will not cause irritation due to the pH in the gastrointestinal tract [24].

#### 3.6.4. DSC

The DSC curves of XH, CE, physical mixture, blank-SLNs, and optimized XH-SLNs have been presented in Figure 6. XH, CE and PF-68 showed sharp melting endothermic peaks at 172.84 °C (A), 72.77 °C (B) and 55.89 °C (C), respectively, which confirmed that they are crystalline in nature. It is pertinent to add here that the thermogram of XH initially showed an endotherm followed by exotherm [54]. The reported melting points of XH is 172 °C [55], CE is 65–77 °C [56] and PF-68 is 52 °C [57]. The LE-80 did not show any endothermic peak that indicated about its amorphous form. However, slight glass transition (T_g_) is observed for LE-80 between 45–60 °C (Figure 6G). In case of blank SLNs and XH-SLNs, a small bifurcated peak was observed in the range of 55 to 65 °C, which indicated the presence of CE and PF-68 in the dispersion of SLNs. However, no such peaks were observed at the melting point of XH in the formulation (D). This indicated that XH was retained in the amorphous form in the SLNs. In addition, the formation of particles in the nanometer range would also have contributed in decreasing the crystallinity of XH. Thus, the outcomes of the DSC study indicated the reason for enhancement in release of XH. The results of DSC of drug, excipients and formulation are shown in Figure 6. To have a better insight, the results of the DSC studies were correlated with PXRD studies of lyophilized SLNs. The loss in crystallinity of XH can be attributed to its complete entrapment/solubility in the milieu of lipids and surfactants.

#### 3.6.5. PXRD

The PXRD of CE exhibited sharp diffraction peaks at 2Ɵ angles of 21.3° and 24.7°. XH exhibited sharp diffraction peaks at 2Ɵ angles of 15.0°, 20.0°, 23.1°, and 26.8°. PF-68 showed sharp diffraction peaks at 20.5° and 26.0°. Figure 7 represents the diffractograms of XH, CE, LE-80, PF-68, XH-SLNs and blank SLNs. The PXRD pattern of PF-68 depicted sharp peaks suggesting their crystalline nature. This complemented the observations of DSC. LE-80 showed a halo pattern and absence of sharp peaks, thus confirming its non-crystalline nature. Similarly, some peaks were observed in the blank SLNs (F) and XH-SLNs (E) at the diffraction angles of CE and PF-68, but the peaks pertaining to XH were completely absent. This indicated complete solubility of XH in the mixture of lipids and surfactants. Thus, the results of PXRD were found in concordance with the results of DSC. 

#### 3.6.6. FE-SEM and HRTEM

A morphological study of XH-SLNs was performed using FE-SEM and HRTEM. The results revealed long, cylindrical and flat images of XH (Figure 8a), indicating its crystalline nature. PF-68 showed flat and irregular reticulated edges, whereas, the images of Compritol E ATO and LE-80 were smooth and waxy in nature, indicating a lipidic nature. The XH-SLNs were amorphous, waxy and spherical in shape, indicating complete loss of crystallinity of XH due to its dispersion in the lipid matrix (Figure 8e). The PS of XH-SLNs was in range of 100 nm scale as per HRTEM results (Figure 8f). The particles exhibited a normal size distribution pattern, which was in agreement with the value obtained during particle size analysis (Section 3.6.1), indicating the absence of aggregation of particles.

### 3.7. In Vitro Drug Release and Release Kinetics

The in vitro drug release of XH from SLNs exhibited slow and sustained release up to 92 h. It was observed that about 28.42% of drug get released up to 12 h followed by 37.2% and 60.25% at 24th and 36th h, respectively. At the end of 92 h, the release of XH from SLNs was found to be 80.56%. This clearly stated that the release pattern of XH from the developed XH-SLNs was continuous and sustained (Figure 9). Interestingly, the naive XH showed a poor release profile owing to its lipophilicity and poor permeability across the membrane [58]. This can be understood by the fact that only 18.40% naive XH was released in 24 h and only 36.54% in 92 h. This indicated that a significant increase (*p* < 0.02) in the release of XH by 2.2-fold was observed upon loading it into SLNs in a sustained manner. This sustained release of XH could be due to the immobilization of XH in the lipids matrix, which hinders the entry of the aqueous phase into the lipid matrix, thereby not allowing sufficient wettability to the XH. [59]. Thus, it was concluded that the SLNs produced at low temperatures and at optimum concentration of the surfactant will not show burst release and drug partitioning into the aqueous phase. Therefore, controlling of drug partitioning into the water phase automatically increases the drug solubility in the lipid phase, which allows the sustained release of the drug without burst release. It also depends on the method of incorporation of the drug into the lipid and also particle size. When compared to large-sized particles, small-sized particles show immediate release of the drug and are also observed with the drug enriched shell model of incorporation. 

#### Release Kinetics of Optimized Formulation

The release patterns of XH-SLNs was evaluated by applying various release kinetics models. These include “Zero-order, First order, Higuchi, Weibull, Hixon Crowell and Krosmeyer-Peppa’s plots and equations”. In case of Weibull, Hixon Crowell and Krosmeyer-Peppa’s models, the value of “R^2^” was observed as less than 0.9, which revealed that the XH’s release patterns did not follow these models. Further, the release pattern of XH-SLNs was applied to Zero order, First order and Higuchi kinetics. The values of “R^2^” was more than 0.9 in the case of all three models; however, the maximum value of 0.975 was observed in the case of First order and Higuchi model. These results inferred that the release pattern was best fitted for First order and Higuchi kinetic models. It may be due to the hydrophobic nature of XH as well as the lipid used in SLNs [60]. Higuchi’s model indicated that the release occurred through diffusion process. The results of the release kinetics of the XH-SLNs are shown in Table 6. 

The sustained release of XH-SLNs was obtained due to the presence of L E-80, which is expected to form a bilayer around the SLNs. This was evident from the study conducted by Hashem Heiati and co-workers in 1996 [59]. According to their findings, the phospholipid present in the emulsion will form a stable bilayer around the lipid core of SLNs, which is responsible for sustained release of XH from the XH-SLNs. Therefore, the increased circulation, residence time and sustained release of XH was attributed to the spread of the LE-80 double layer upon the solid lipid core of XH-SLNs. Furthermore, CE, which is a solid lipid, also played a role in providing sustained release to XH.

### 3.8. Cell Permeability Studies

The transport of XH through the intestinal membrane by SLNs was evaluated using in vitro cell line permeability. At 5 h, the release of XH from XH-SLNs was found to be 6.2 ± 0.98 nmol, whereas naive XH showed only 1.34 ± 0.087 nmol permeation. Hence, the permeation of XH from XH-SLNs was about 4.62 folds higher (A-B) than naive XH and 0.31-fold lower drug excretion (B-A) than naive XH (Figure 10). For understanding the bioavailability of the drug, permeability is an essential characteristic because it is directly related to the availability of the drug in systemic circulation [61].

### 3.9. In Vivo Pharmacokinetic Study

The non-compartmental pharmacokinetic technique is more adaptable since it does not rely on a compartmental pharmacokinetic model and generates accurate results. This approach is extensively utilized in bioequivalence studies because it has the benefit of requiring few assumptions about the data-gathering procedure and allowing for extremely organized data collection [62]. Further, the results revealed about 1.07-fold enhancement in C_max_ upon loading XH into SLNs as its value was 20,730 ng/mL for XH that got enhanced to 22,350 ng/mL. This indicated higher absorption of XH loaded in SLNs [61,63]. Similarly, the results of AUC_0-t_ revealed about 4.7-fold enhancement in its result for XH upon loading into SLNs as that of naive XH (Table 7, Figure 11). The value of AUC_0-t_ was 71,360 ng/mL h for naive XH that got enhanced up to 341,900 ng/mL h upon loading it in to SLNs. Higher AUC is indicative of enhancement in oral bioavailability of the drug [64]. The presence of lipids and surfactants cause the formation of micelles containing the drug in nanometer range that may be rapidly absorbed in systemic circulation through the GI mucosa as well as lymphatic route without producing any back reflux, thereby leading to enhanced oral bioavailability. Furthermore, the presence of lipid in the droplet would offer slow release of the drug, thus producing a larger AUC [62]. In addition, SLNs have been reported to play an active role in enhancing the oral uptake of drugs after bringing it in the solubilized form into the GIT and subsequent formation of micelles. The lipases present in the GIT break the triglycerides into the surface-active mono- and diacylglycerols that further stimulate the secretion of bile salt endogenously [65,66,67]. Overall, it may help in enhancing oral bioavailability [62,63]. 

In this study, it was also observed that t_1/2_ was increased from 1.97 h (XH) to 12.76 (XH-SLNs) (6.47-fold) and MRT increased from 3.583 h (XH) to 21.97 h (XH-SLNs) (6.13-fold). The higher value of t_1/2_ and MRT indicated the prolongation of circulation time of the drug in systemic circulation. The relative bioavailability was found to be 4791% for XH loaded in SLNs, and the relative bioavailability of naive XH was 20.80%. This indicated about 230.33-folds enhancement in bioavailability upon loading XH into SLN as compared to naive XH [61,63,64]. Hence, the results of the release studies were found in concordance with pharmacokinetic studies.

### 3.10. Cytotoxicity Study on PC-3 Cell Lines

The PC-3, LNCap and DU145 are the most commonly used prostate cancer cell lines, but they have different characteristics. The most important advantages of PC-3 cell lines among other prostate cancer cell lines is that they are androgen independent, divide normally in androgen deprived media and their xenografts grow rapidly and they are also more invasive in comparison to other cell lines [68]. As a result, the PC-3 cell lines have been selected for the present research work. The importance of PC-3 cell lines can be understood by their use in various research work done by many researchers, as described previously in the introduction section. 

The percentage cell inhibition was tested on PC-3 cancer cells for four different intervals, i.e., at 6, 12, 24 and 48 h at six different concentrations. Camptothecin was used as a positive control to compare the percentage of inhibition between naive XH and XH SLNs. The cell inhibition percentage was noted in a dose- and time-dependent manner. Among all four groups, the naive XH has shown the maximum percentage of cell inhibition at the end of 48 h. This finding is consistent with earlier research on the impact of XH on prostate cancer cell lines [69,70,71]. Incidentally, the placebo has also shown some extent of percentage cell inhibition, probably due to the presence of surfactants used in the formulation. It is important to note that a significant (*p* < 0.05) reduction in inhibitory activity was observed in the case of XH-SLNs as that of naive XH in the first 12 h. This was due to slow and sustained release of XH from the formulation, as observed during the release studies. In contrast, the inhibitory activity of the XH-SLNs were enhanced in the later stage in the case of XH-SLNs at 24 h and 48 h due to sufficient availability of free drugs for interaction with the PC-3 cells leading to cellular toxicity. This can be understood by looking at the results of percentage cell inhibition by naive XH and XH-SLNs at 24 h and 48 h, wherein there is no significant difference (*p* > 0.05) in their results. The increase in cytotoxicity in a time-dependent manner suggests that the release of XH is in a sustained pattern from the lipid matrix of XH-SLNs [20]. The overall study indicated that XH possess good anticancer activity as such against PC-3 cancer cells, and this effect can be prolonged for a longer time upon loading into SLNs. The results of PC-3 cell line study was represented in Figure 12.

## 4. Conclusions

Optimized XH-SLNs were developed by using high shear-homogenization/ultrasonication method, and characterizations such as, PS, PDI, ZP, DSC, PXRD, FE-SEM, and HRTEM were performed. The in vitro release kinetics for optimized XH-SLNs were studied and compared with the naive XH. The release profile of XH-SLNs was best fitted with first order kinetics, Krosmeyer-Peppas, and Higuchi models (r^2^ > 0.9). Therefore, based on these release kinetic results, XH-SLNs developed were suitable for assessing lymphatic transport pathway for enhancing oral bioavailability of XH. Lower PS, narrow PDI, and optimum ZP were observed, which indicated the physical stability of developed XH-SLNs. The surfactants used in the development of formulation was non-ionic (L E-80 and PF-68), which are safer physiologically. The optimum levels of these surfactants may be responsible for slow, extended and sustained release rather than burst release of XH, which is mainly due to formation of LE-80 coat around the solid lipid (CE). DSC and PXRD studies revealed the amorphous nature of XH-SLNs formulation. CE used as solid lipid, which is responsible for protection of SLN in gastric environment, even though there is partial lipolysis in the acidic environment that leads to formation of micelles, followed by cellular uptake. Therefore, the loaded drug can be protected in the gastric environment from biotransformation and safely transported into the distribution phase with an improved pharmacokinetic profile. The morphological evaluation was done by FE-SEM and HRTEM, revealing the spherical morphology of XH-SLNs, which is responsible for prevention of agglomeration of solidified particles. The developed XH-SLNs are the most suitable carrier system to improve the oral bioavailability of XH, which was its major limitation. The developed XH-SLNs were also evaluated for their anticancer effect using PC-3 cell lines by taking camptothecin as a positive control. XH-SLNs showed a noticeable anticancer effect on PC-3 cell lines in a sustained pattern. The overall outcomes of the study showed that XH can be further explored for its anticancer potential in suitable animal model. Furthermore, the formulation can be checked for its storage stability and scaling up aspects in future studies. Nevertheless, it offers a novel anticancer therapeutic from natural products that may be the encapsulation of natural compounds into SLNs. As XH acts through multiple pathways, it can be used as a nutraceutical for providing antioxidant, anticancer, anti-inflammatory as well as antidiabetic effects. Hence, the developed XH-SLNs can be explored further for checking their efficacy in treating other types of cancers apart from prostate cancer as well as other diseases such as diabetes, cardiovascular diseases and neurodegenerative diseases. Furthermore, the successful outcome of this study also provides an insight that the developed formulation can be tested on suitable animal models of cancer.

## Figures and Tables

**Figure 1 pharmaceutics-14-02403-f001:**
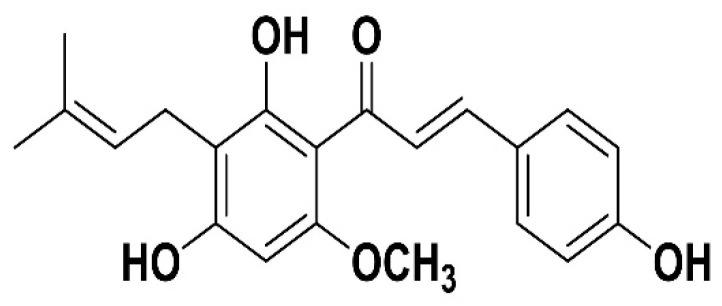
Structure of Xanthohumol.

**Figure 2 pharmaceutics-14-02403-f002:**
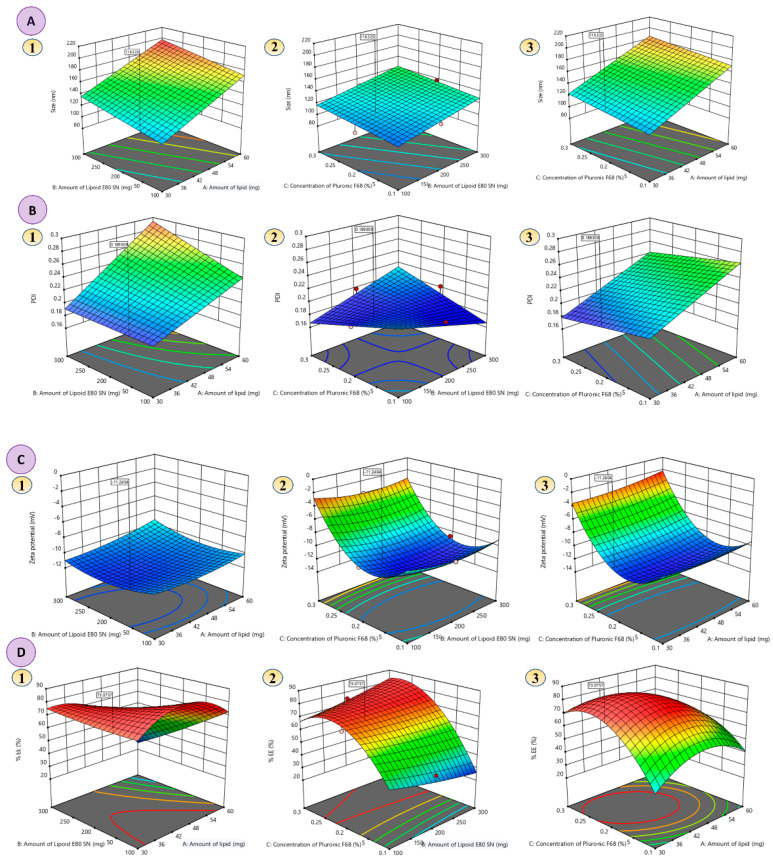
(**A**–**D**) 3D-response plots representing the effects of CMAs on (**A**) PS; (Effects of amount of lipid, amount of lipoid and concentration of PF-68 (1–3), (**B**) PDI; (Effects of amount of lipid, amount of lipoid and concentration of PF-68 (1–3) (**C**) ZP; (Effects of amount of lipid, amount of lipoid and concentration of PF-68 (1–3) and (**D**) %EE; (Effects of amount of lipid, amount of lipoid and concentration of PF-68 (1–3).

**Figure 3 pharmaceutics-14-02403-f003:**
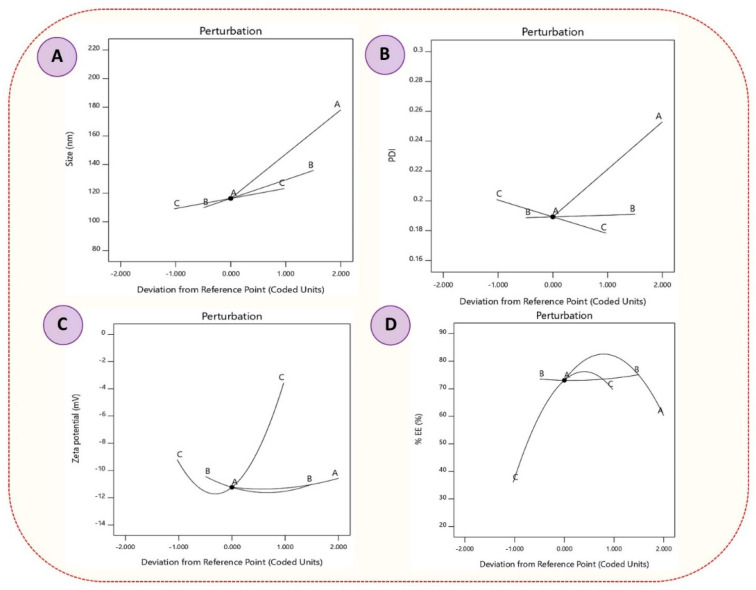
(**A**–**D**) Perturbation plots (**A**) PS, (**B**) PDI, (**C**) ZP and (**D**) %EE.

**Figure 4 pharmaceutics-14-02403-f004:**
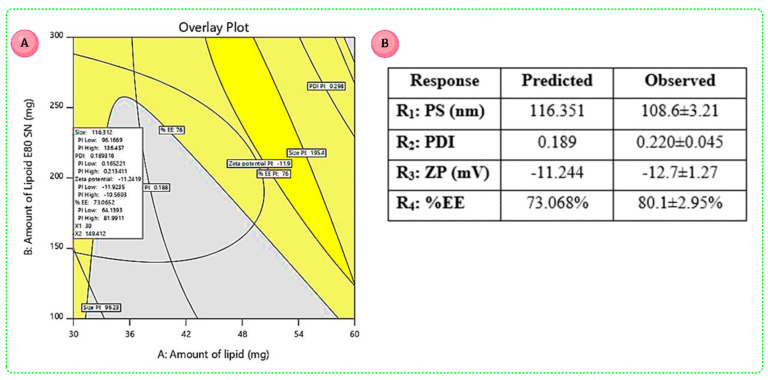
(**A**) Overlay plot representing area of optimized formula for XH-SLNs. (**B**) Predicted and observed values of the optimized batch of XH-SLNs.

**Figure 5 pharmaceutics-14-02403-f005:**
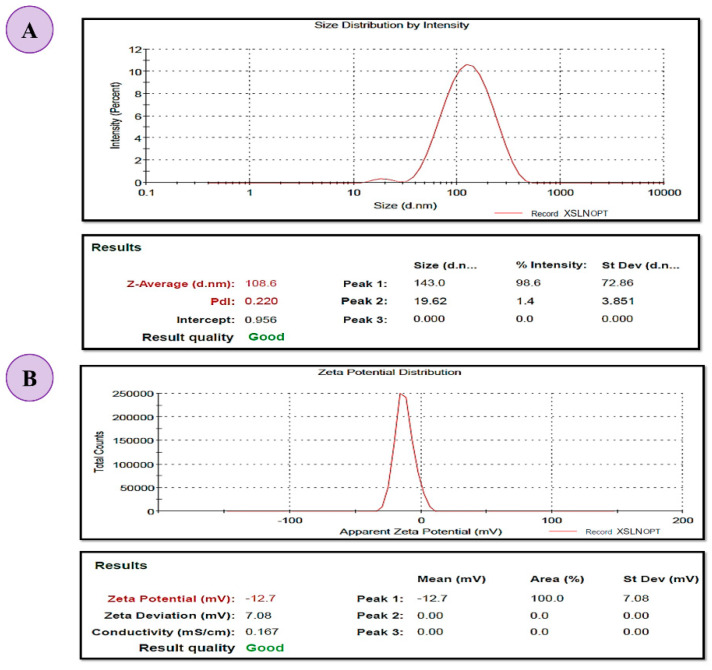
Representing (**A**) PS and PDI, and (**B**) ZP of the optimized XH-SLNs.

**Figure 6 pharmaceutics-14-02403-f006:**
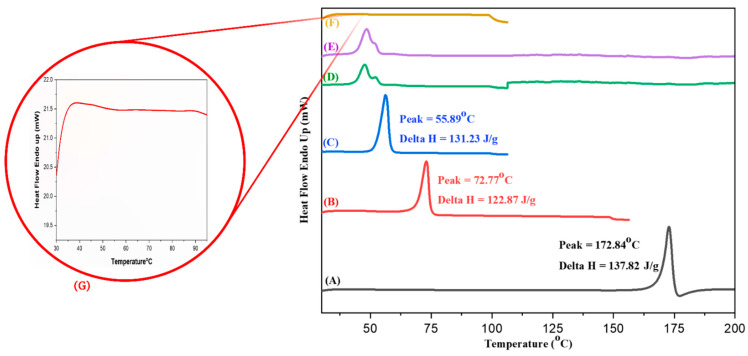
Comparative DSC curves of (**A**) XH, (**B**) CE, (**C**) Pluronic F-68 (PF-68), (**D**) XH-SLNs, (**E**) Blank-SLNs and (**F**) LIPOID E 80 SN (LE-80). (**G**) Zoom image of DSC of LE-80.

**Figure 7 pharmaceutics-14-02403-f007:**
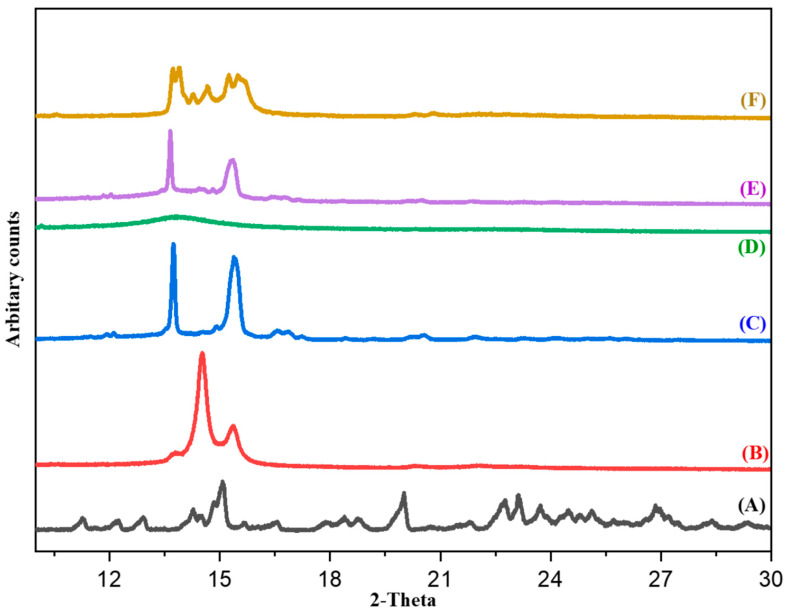
Representing the PXRD patterns of (**A**) XH, (**B**) CE, (**C**) Pluronic F-68 (PF-68), (**D**) Lipoid E 80SN (LE-80), (**E**) XH-SLNs and (**F**) Blank-SLNs.

**Figure 8 pharmaceutics-14-02403-f008:**
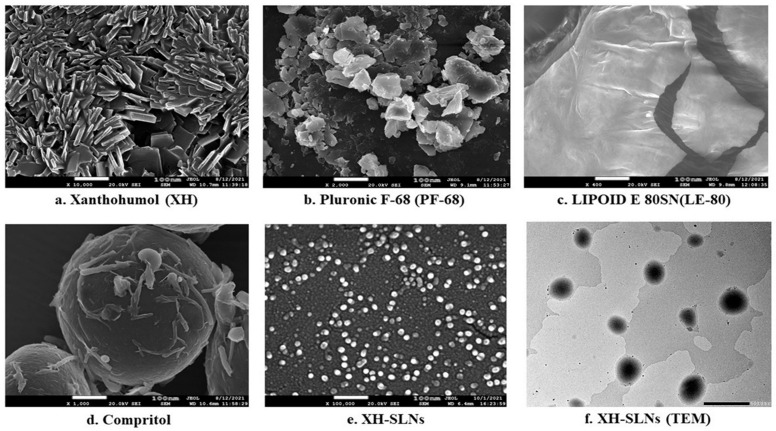
FE-SEM images of (**a**) Xanthohumol; (**b**) Pluronic F-68; (**c**) Lipoid E 80SN; (**d**) CE; (**e**) XH-SLNs and HRTEM images of (**f**) XH-SLNs.

**Figure 9 pharmaceutics-14-02403-f009:**
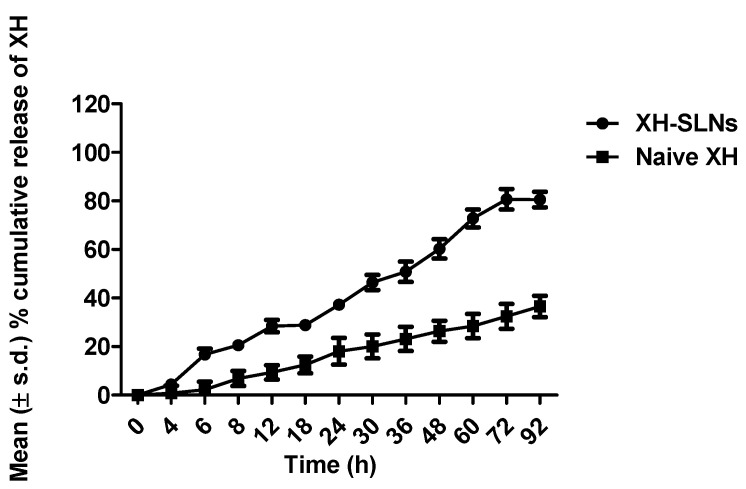
Release profile of naive XH and XH-SLNs.

**Figure 10 pharmaceutics-14-02403-f010:**
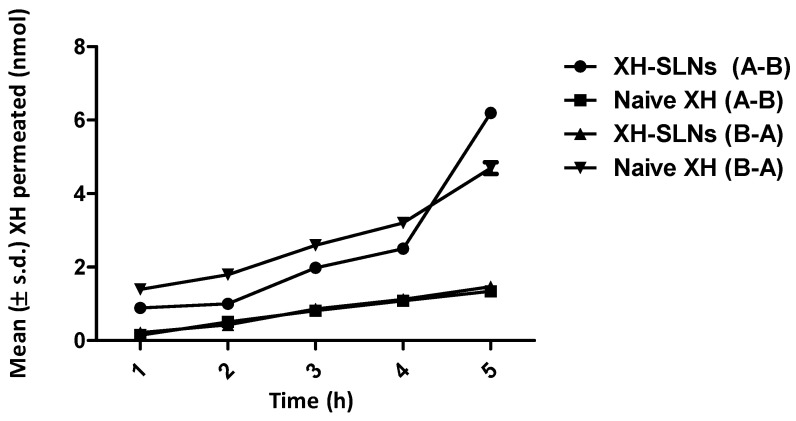
Permeability data of XH in Caco2 cell monolayer (number of replicates = 3).

**Figure 11 pharmaceutics-14-02403-f011:**
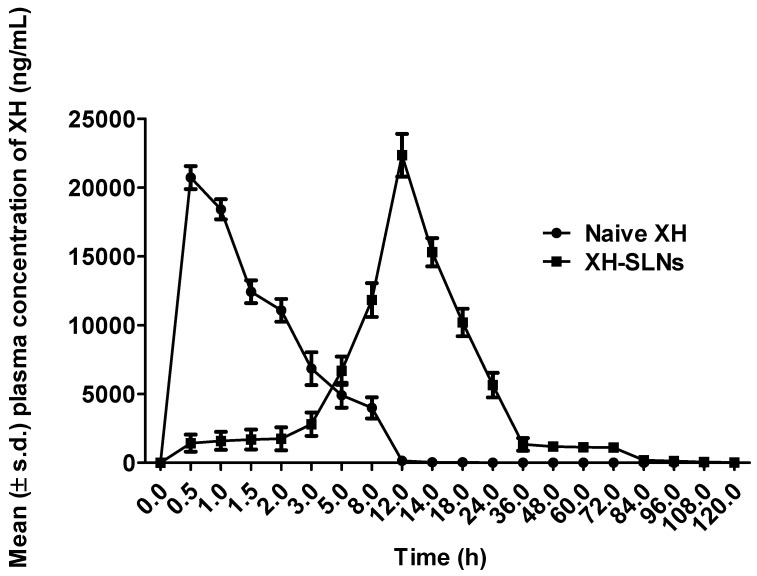
Plasma drug concentration-time profile of naive XH and XH-SLNs.

**Figure 12 pharmaceutics-14-02403-f012:**
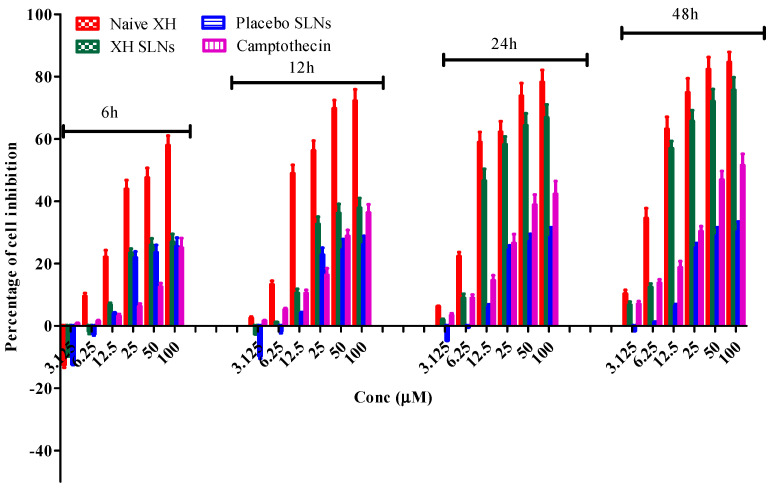
Results of PC-3 cell line % cell inhibition by naive XH, XH-SLNs, placebo SLNs and camptothecin (positive control).

**Table 1 pharmaceutics-14-02403-t001:** Solubility of XH in various solid lipids.

Lipid	Drug-Lipid Ratio (mg)				
	1:2	1:3	1:4	1:5	1:6
GMS	⁎	⁑	⁂	⁂	⁂
Stearic acid	⁎	⁎	⁑	⁂	⁂
Palmitic acid	⁎	⁎	⁑	⁑	⁂
Carnauba wax	⁎	⁎	⁎	⁎	⁑
Compritol E	⁎	⁂	⁂	⁂	⁂
Cetyl alcohol	⁎	⁎	⁎	⁑	⁑
MONEGYL T18	⁎	⁎	⁎	⁑	⁑
Precirol ATO5	⁎	⁑	⁂	⁂	⁂

⁎ Not clear, ⁑ Turbid, ⁂ clear.

**Table 2 pharmaceutics-14-02403-t002:** Formulations of XH-SLNs with various surfactants and co-surfactants.

Formulation	Solid Lipid	Surfactant (2%)	Co-Surfactant (1%)	PS (nm)	PDI	ZP (mV)
XH-SLNs	Compritol E ATO	Tween 80	-	409.90 ± 0.90	0.65 ± 0.08	−12.70 ± 2.85
XH-SLNs		Tween 80	Pluronic F-68	198.00 ± 2.08	0.46 ± 0.04	−30.60 ± 1.74
XH-SLNs		Monemul-20	-	662.50 ± 6.40	0.57 ± 0.02	−3.98 ± 3.82
XH-SLNs		Monemul-20	Pluronic F-68	313.30 ± 0.61	0.61 ± 1.4	−17.40 ± 3.72
XH-SLNs		LIPOID S 75	-	684.90 ± 2.62	0.67 ± 0.04	−8.70 ± 2.32
XH-SLNs		LIPOID S 75	Pluronic F-68	581.20 ± 1.85	0.28 ± 0.08	−20.50 ± 4.25
XH-SLNs		LIPOID E 80 SN	-	152.20 ± 3.91	0.22 ± 0.00	−18.50 ± 1.02
XH-SLNs		LIPOID E 80 SN	Pluronic F-68	118.20 ± 0.14	0.17 ± 0.07	−12.60 ± 2.84
XH-SLNs		PHOSPHOLIPON 90 H	-	416.20 ± 1.56	0.57 ± 0.02	−24.30 ± 0.12
XH-SLNs		PHOSPHOLIPON 90 H	Pluronic F-68	480.60 ± 2.11	0.42 ± 0.07	−8.06 ± 3.71

**Table 3 pharmaceutics-14-02403-t003:** Preliminary screening of solid lipid, surfactant and co-surfactant.

Batch	Solid Lipid	Surfactant (2% *w*/*v*) + Co-surfactant (2% *v*/*v*)	PS (nm)	PDI	ZP (mV)	%EE
**1**	Precirol ATO 5	LIPOID E 80 SN + Pluronic F-68	152.10 ± 3.28	0.37 ± 0.004	−12.60 ± 1.64	68.40% ± 3.96
**2**	GMS		179.80 ± 2.14	0.24 ± 0.85	−13.50 ± 3.58	62.20% ± 3.45
**3**	CE		135.70 ± 3.65	0.28 ± 0.32	−12.70 ± 3.13	74.60% ± 2.74

**Table 4 pharmaceutics-14-02403-t004:** Preliminary screening of speed and time of homogenization.

Batch	Time of Homogenization (min)	Speed of Homogenization (rpm)	PS	PDI	%EE
1	15	6000	351.60 ± 3.43	0.55 ± 0.08	70.60 ± 2.15%
2	20	6000	133.90 ± 5.47	0.41 ± 0.05	71.90 ± 2.58%
3	30	6000	154.50 ± 3.84	0.62 ± 0.02	68.20 ± 3.07%
4	20	8000	140.60 ± 4.89	0.26 ± 0.03	78.50 ± 3.43%
5	20	10,000	259.40 ± 4.28	0.68 ± 0.05	68.80 ± 2.76%

**Table 5 pharmaceutics-14-02403-t005:** BBD table indicating experimental observed responses.

S. NO	A: Amount of Lipid (mg)	B: Amount of Lipoid E80SN (mg)	C: Concentration of Pluronic F68 (%)	R_1_: PS (nm)	R_2_: PDI	R_3_: ZP (mV)	R_4_: %EE
XH-SLNs_1_ *	45	200	0.2	153.54 ± 5.28	0.23 ± 0.07	−11.50 ± 1.85	75.12 ± 2.74
XH-SLNs_2_	30	200	0.1	110.56 ± 4.37	0.19 ± 0.03	−9.80 ± 2.36	37.20 ± 3.24
XH-SLNs_3_	45	300	0.1	162.86 ± 3.53	0.20 ± 0.05	−9.03 ± 3.45	24.00 ± 2.78
XH-SLNs_4_	60	300	0.2	179.20 ± 6.79	0.29 ± 0.06	−9.36 ± 4.85	28.80 ± 1.29
XH-SLNs_5_ *	45	200	0.2	156.22 ± 2.85	0.21 ± 0.01	−11.50 ± 3.25	75.90 ± 3.11
XH-SLNs_6_	60	200	0.1	178.54 ± 1.83	0.27 ± 0.08	−9.35 ± 2.01	23.80 ± 1.05
XH-SLNs_7_	45	100	0.1	136.23 ± 6.67	0.24 ± 0.07	−8.73 ± 1.75	62.40 ± 3.12
XH-SLNs_8_	45	300	0.3	172.24 ± 9.75	0.25 ± 0.04	−2.76 ± 3.86	67.00 ± 8.65
XH-SLNs_9_	30	300	0.2	137.39 ± 1.58	0.20 ± 0.01	−10.80 ± 6.45	71.20 ± 1.78
XH-SLNs_10_ *	45	200	0.2	153.28 ± 3.75	0.21 ± 0.06	−11.30 ± 8.10	75.50 ± 1.98
XH-SLNs_11_ *	45	200	0.2	181.30 ± 4.78	0.22 ± 0.07	−11.90 ± 4.52	75.80 ± 2.86
XH-SLNs_12_	30	200	0.3	128.26 ± 7.45	0.19 ± 0.01	−3.92 ± 3.12	75.00 ± 3.48
XH-SLNs_13_	45	100	0.3	148.54 ± 1.24	0.19 ± 0.08	−2.56 ± 3.56	74.60 ± 2.98
XH-SLNs_14_ *	45	200	0.2	155.26 ± 3.22	0.22 ± 0.09	−11.70 ± 2.45	75.70 ± 1.43
XH-SLNs_15_	30	100	0.2	96.23 ± 3.72	0.18 ± 0.03	−10.60 ± 4.85	70.00 ± 2.93
XH-SLN_16_	60	100	0.2	166.78 ± 7.48	0.23 ± 0.05	−10.50 ± 7.25	76.00 ± 1.86
XH-SLN_17_	60	200	0.3	195.40 ± 4.36	0.28 ± 0.05	−1.98 ± 4.52	26.40 ± 3.75

* indicates center points.

**Table 6 pharmaceutics-14-02403-t006:** Release kinetics of optimized formulation.

Model	Parameters				
	R^2^	K	RMSE	AIC	BIC
Zero order	0.936	1.4	7.13	9.59	9.72
First order	0.975	0.008	4.84	2.14	2.15
Higuchi	0.975	2.72	3.20	1.37	1.39
Weibull	0.813	4.93	3.98	7.96	8.09
Hixon Crowell	0.722	3.33	1.77	1.21	1.22
Krosmeyer Peppas	0.890	1.56	2.34	1.29	1.30

**Table 7 pharmaceutics-14-02403-t007:** Representing the pharmacokinetic data of XH and XH-SLNs.

Parameter	Unit	Value (Naive XH)	Value (XH-SLNs)
t_1/2_	h	1.975	12.76
t_max_	h	0.5	12
C_max_	ng/mL	20,730	22,350
AUC_0-t_	ng/mL h	71,360	341,900
AUC_0-inf_obs_	ng/mL h	71,390	342,900
MRT_0-inf_obs_	h	3.58	21.97

## Data Availability

Not applicable.

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
