# Peer review of "Quality by Design Based Formulation of Xanthohumol Loaded Solid Lipid Nanoparticles with Improved Bioavailability and Anticancer Effect against PC-3 Cells"

_pharmaceutics, 2022, doi:10.3390/pharmaceutics14112403_

Round 1

Reviewer 1 Report

The work presented by Harish et al, on quality by design based formulation of Xanthohumol (XH) loaded solid lipid nanoparticles (SLNs) with improved bioavailability and anticancer effect against PC-3 cells is a novel work as no SLNs of xanthohumol are reported to treat prostate cancer. In addition, cancer is a deadly disease that is mushrooming very fast across the globe. Developing a nano-formulation for a phytoconstituent for treating cancer provides a holistic way to treat the disease as it is safe as compared to synthetic drug and offers the opportunity for the scientists who are working in the area of natural product research. In the present study XH loaded SLNs were developed using quality by design approach that has helped in understanding of formulation variables affecting development of SLNs and created an optimized design space to get the reproducible formulation quality. The developed formulation has been well characterized using FESEM, HRTEM, PXRD and DSC studies. The release of formulation was enhanced in a sustained manner leading to long therapeutic effect of formulation. This was confirmed by the authors by the PC-3 cell lines studies. The work is well organized and the proposed hypothesis is well proven through obtained results. It can be accepted after minor revision.

1. Introduction: Authors are encouraged to discuss the outcomes of some previous studies wherein SLNs have successfully being used to treat prostate cancer.

2. The language of manuscript should be thoroughly check and corrected wherever required.

3. Some more discussion should be added in the results of PC-3 cell line studies based on available literature about the suitability of this cell line for prostate cancer over other cell lines.

4. Conclusion: Authors should end with some future perspectives that how they think that their formulation should be taken for future research.

Author Response

Respected sir, Please see the attachment

Reviewer 2 Report

Manuscript entitled “Quality by design based formulation of Xanthohumol loaded solid lipid nanoparticles with improved bioavailability and anticancer effect against PC-3 cells” contains original data on the formulation and characterization of SLNs containing xanthohumol. The manuscript is based on huge amount of work, using state of the art techniques for the preparation and characterization of the nanoparticles, however the presentation and description of results must be improved as well as the English of the text should be improved. 

The comments are the following: 

Why “naïve” is used instead of “naive” in the text, while on the graphs “naive” can be found. 

2.2.2. Determination of XH solubility in various solid lipids 

Initially 10 mg of lipid was taken in a beaker and melted by heating 10°C above its melting point. XH (10 mg) was accurately weighed and added to the molten lipid with constant stirring using magnetic stirrer” - How can 20 mg of material be stirred by a magnetic stirrer? Their quantity is too small for any kind of magnetic stirrer.  

2.2.7. Lyophilization of optimized XH-SLNsnot described in detail and correctly. 

2.3.6. Field Emission Scanning Electron Microscopy (FESEM) 

The stub was then coated with gold tothickness of 200 to 500 microns using a gold sputter module in a high vacuum evaporator in an argon environment. - Please explain that how the nanoparticles were coated with gold at 200-500 microns thickness. In this case the nanoparticles were fully covered with gold an could not be detected. 

2.3.10. Cell permeability study 

The naïve XH solution was prepared by dissolving it in methanol. The XH-SLNs and naïve XH solution were added to the side A and side B of the cell inserts respectively. - XH solution was still in methanol on the cells? It is toxic.  

2.3.11.2. Pharmacokinetic StudyWhat was the standard? 

Figure 2, 3, 4 axis titles are too small.  

Figure 5. - The quality of Figure 5 is not suitable for reading and publication. 

3.6.3. - Based on only pH values, it cannot be concluded, that XH-SLNs will not cause any irritation in the GI tract.  

Fig. 8. Scale bar should be added onto the images with the correct size. 

3.8. Units are not written in the text.  

Figure 10. - Number of replicates should be given and S.D. should be presented on the graphs. 

Abstract – “cancer study” and “anticancer effect” are not suitable terms for in vitro studies on cells.

Author Response

(The authors gave the same response as above.)

Reviewer 3 Report

The manuscript “Quality by design based formulation of Xanthohumol loaded 2 solid lipid nanoparticles with improved bioavailability and an-3 ticancer effect against PC-3 cells” presents an interetsedting tudy involving the development of Xanthohumol loaded solid lipid nanoparticles. Authors have used QbD for formulation optimization and subsequently performed thorough characterization. The overall study looks well planned, and interesting for publication. However, the writing part needs to be strengthened. The specific comments are provided below:

Major comments:

1.      ‘Materials and Methods’ and ‘Results and Discussion’ sections need to be concise. Repetitive information has been provided at multiple places.

2.      The MS needs a thorough proof correction to remove multiple grammatical errors.

3.      The interpretation of DSC and PXRD lacks clarity. Authors are suggested to re-look at the data and re-write these two sections.

4.      Section 2.2.7: The procedure of lyophilization looks incomplete. It is almost impossible to lyophilize at -80°C within 48 hours. The temperature is too low to favor sublimation

5.      Figure 7 should be cropped to zoom the region of interest (i.e.. 5 to 40 ° 2-theta)

6.      Line 694- 705: The explanation is too generic and should be concised

7.      Line 748- 760: The explanation is too generic and should be concised

Minor comments:

1.      Line 132- Reference for the previously validated method has not been provided

2.      Section 2.2.2- Solubility is a function of temperature. The protocol used suggest that the temperature of solubility assessment was not constant for all the lipids. Authors should discuss this in Results and discussion.

3.      Line 709-Line 719- The write up is casually written. It should be concised and re-written

4.      Line 770: How is the ‘relative bioavailability’ calculated? Is this the correct term to be used here?

5.      Line 748- AUC is almost identical for the two formulations. The results should be explained accordingly.

Author Response

(The authors gave the same response as above.)

Round 2

Reviewer 2 Report

The manuscript was improved significantly, however some of the questionable parts were not improved. My comments are the following:

“naïve” is still used in the text.

Please give the type of freeze dryer.

Fig 2 -please enlarge the font size on the figure.

Figure 5 –please put part B under part A and enlarge both.

3.6.3 – the conclusion is still wrong. Not only the pH value could cause irritation. The statement „will not cause any irritation to the oral cavity as well as gastrointestinal tract upon oral administration” is incorrect. You can say only that it „will not cause irritation due to the pH”.

Fig. 8. Please give the size of each scale bar.

Reviewer 3 Report

The authors have attempted to address the comments. However, some replies are not satisfactory. I have elaborated the comments so authors can update it specifically:

Major comments:

1.      Authors have attempted to concise the ‘Results and Discussion’. However, no attempts have been made to concise ‘Materials and Methods’. The following sections should be concise.

a.      2.2.2

b.      2.2.5.2

c.      2.3.11.2

2.      The interpretation of DSC and PXRD still needs improvement. Authors are suggested to re-look at the data and re-write these two sections. I have elaborated some points below.

a.      DSC

                                                    i.     The number of significant figures should be corrected based upon the accuracy of the instrument. For most instruments, quantitative values are not be more than 1-2 decimal place for temperature and enthalpy.

                                                  ii.     The usual convention currently followed for the DSC heating curve globally is ‘exo up’. I suggest plotting the heating curve accordingly.

                                                 iii.     The values on the y-axis should be removed as it is an overlay. The axis label should be ‘Heat flow (arbitrary units)’

                                                 iv.     The observed melting temperatures and enthalpies for XH, CE and PF-68 should be compared with literature

                                                   v.     The transition observed for XH is endotherm followed by an exotherm. This has not been addressed/pointed out. Are the any reports that have described thorough thermal analysis of this component.

                                                 vi.     LE-80 did not show an endothermic peak, thus suggesting lack of crystallinity. This was further complemented by XRD where a halo pattern was observed. I would suggest to zoom in the heating curve to observe for any ‘glass transition’ event.

                                               vii.     The ‘delta H’ values of 65.9966 and 63.5064 in curves D and E should be removed as it appears to be noise rather than an endotherm.

                                              viii.     Based on the DSC heating curve, the authors stated that XH was ‘completely soluble’ in SLNs. The term ‘soluble’ may not be appropriate here. Rather, the authors can state the XH was retained in the amorphous form in the SLN.

                                                 ix.     Line 684 to 688: ‘ This indicated…..micelles of SLNs’ should be deleted.

b.      XRD

                                                    i.     It is suggested to report the 2-theta values upto 1 decimal only, considering the accuracy and precision of the instrument

                                                  ii.     In the PXRD section, it would be more interesting to compare the DSC and PXRD results. For example, “The XRD pattern of PF-68 depicted sharp peaks suggesting its crystalline nature. This complemented the observations of DSC”.

                                                 iii.     It should be mentioned that ‘LE-80’ showed a halo pattern and absence of sharp peaks, thus confirming its non-crystalline nature.

                                                 iv.     The diffraction pattern of blank SLN and XH loaded SLN should be cautiously explained. Multiple peaks are observed in the curve F. However, these are absent in E. This should be explained in detail. This is uncommon.  

                                                   v.     I suggest removing the casual discussion mentioned in lines 701- 706’ i.e., ‘Similarly, ………results of DSC’

                                                 vi.     Lines 285-288 ‘Powder…formulation’ should be deleted.

                                               vii.     “PXRD spectra” this term should be corrected as ‘PXRD diffractogram’ throughout the manuscript

                                              viii.     Line- 292- The step size of XRD has been given in °C. This is incorrect.

3.      Section 2.2.7: The procedure of lyophilization still needs improvement

a.      Lines 233- 237 “Lyophilization …. Of SLN [29]’ should be deleted.

b.      What was the make of the instrument?

c.      Was it a shelf lyophilizer or a flask?

d.      What were the conditions for secondary drying?

e.      How was the vacuum broken, using ambient air or dry nitrogen?

f.       What were the conditions at the time of unloading the sample?

g.      What are the references [30-32] for?

4.      Figure 7

a.      Numbers on y-axis should be removed as it is an overlay

b.      The y-axis label should be ‘arbitrary counts’

c.      X-axis label should be ‘2-theta’

5.      Method validation data should be provided in supplementary information as it will be a useful addition for the researchers.
